



# Refining physical aspects of soil quality and soil health when exploring the effects of soil degradation and climate change on biomass production: an Italian case study.

Antonello Bonfante[1], Fabio Terribile[2,3], Johan Bouma[4]

[1]Institute for Mediterranean Agricultural and Forest Systems - CNR-ISAFOM, Ercolano, Italy
[2]University of Naples Federico II, Department of Agriculture, Portici, (NA), Italy
[3]University of Naples Federico II, CRISP Interdepartmental Research Centre
[4]Em. Prof. Soils Science, Wageningen University, The Netherlands

*Correspondence to*: Antonello Bonfante (antonello.bonfante@cnr.it)

**Abstract.** This study is restriced to soil physical aspects of soil quality and - health with the objective to define procedures with worldwide rather than only regional applicability, reflecting modern developments in soil physical research and focusing on important questions regarding possible effects of soil degradation and climate change. In contrast to water and air, soils cannot, even after much research, be characterized by a universally accepted quality definition and this hampers the internal and external communication process. Soil quality expresses the capacity of the soil to function. Biomass production is a primary function, next to filtering and organic matter accumulation, and can be modeled with soil-water-plant-atmosphere simulation models, as used in the agronomic yield-gap program that defines potential yields (Yp) for any location on earth determined by radiation, temperature and standardized crop characteristics, assuming adequate water and nutrient supply and lack of pests and diseases. The water-limited yield (Yw) reflects, in addition, the often limited water availability at a particular location. Real yields (Ya) can be considered in relation to Yw to indicate yield gaps, to be expressed in terms of the indicator: (Ya/Yw) x 100. Soil data to calculate Yw for a given soil type (the genoform) should consist of a range of soil properties as a function of past management (various phenoforms) rather than as a single "representative" dataset. This way a Yw-based soil-characteristic soil quality range is defined, based on semi-permanent soil properties. In this study effects of subsoil compaction, overland flow following surface compaction and erosion were simulated for six soil series in the Destre Sele area in Italy, including effects of climate change. Recent proposals consider soil health, which appeals more to people than soil quality and is now defined by seperate soil physical, -chemical and – biological indicators. Focusing on the soil function biomass production, physical soil health at a given time of a given type of soil can be expressed as a point (defined by a measured Ya) on the defined soil quality range for that particular type of soil, thereby defining the seriousness of the problem and the scope for improvement. The six soils showed different behavior following the three types of land degradation and projected climate change up to the year 2100. Effects are expected to be



major as reductions of biomass production of up to 50% appear likely. Rather than consider soil physical, chemical and
biological indicators seperately, as proposed now for soil health, a sequential procedure is suggested logically linking the
seperate procedures.
**1.  Introduction**
The concept of Soil Health has been proposed to communicate the importance of soils to stakeholders and policy makers
(Moebius-Clune et al., 2016). This follows a large body of research on soil quality, recently reviewed by Bünemann et al.,
(2018). The latter conclude that research so far has hardly involved farmers and other stakeholders, consultants and
agricultural advisors. This may explain why there are as yet no widely accepted, operational soil quality indicators in
contrast to quality indicators for water and air which are even formalised into specific laws (e.g. EU Water Framework
Directive). This severely hampers effective communication of the importance of soils which is increasingly important to
create broad awareness about the devastating effects of widespread soil degradation. New soil health initiatives, expanding
the existing soil quality discours, deserve therefore to be supported. A National Soil Health Institute has been established
in the USA ( www.soilhealthinstitute.org) and Cornell University has published a guide for its comprehensive assesment
after several years of experimentation (Mobius-Clune et al, 2016). Soil health is defined as:"*the continued capacity of the*
*soil to function as a vital living ecosystem that sustains plants, animals and humans"*(NRCS, 2012). Confining attention in
this paper to soil physical conditions, the Cornell assessment scheme (Moebius-Clune et.al, 2016) distinguishes three soil
physical parameters: wet aggregate stability, surface and subsurface hardness to be characterized by penetrometers and the
available water capacity (AWC: water held between 1/3 and 15 bar). The National Soil Health Institute reports 19 soil
health parameters, including 5 soil physical ones: water-stable aggregation, penetration resistance, bulk density, AWC and
infiltration rate.
Techniques to determine aggregate stability and penotrometer resistance have been introduced many years ago (e.g. Kemper
and Chepil, 1965; Lowery, 1986; Shaw et al., 1943). Aggregate stability is a relatively static feature as compared with soil
temperature and moisture content with drawbacks in terms of (1) lack of uniform applied methodology (e.g. Almajmaie et
al., 2017), (2) the inability of dry and wet sieving protocols to discriminate between management practices and soil
properties (Le Bissonnais, 1996; Pulido Moncada et al., 2013) and above all: (3) the mechanical work applied during dry
sieving is basically not experienced in real field conditions (Díaz-Zorita et al., 2002). Measured Penetrometer resistances
are known to be quite variable because of different modes of handling in practice and seasonal variation. Finally, the AWC
is a static characteristic based on fixed values for "field capacity" and "wilting point" that don't correspond with field
conditions in most soils (e.g. Bouma, 2018).
These drawbacks must be considered when suggesting the introduction for general use as physical soil health indicators.
More recent developments in soil physics may offer alternative approaches, to be explored in this paper, that are more in
line with the dynamic behavior of soils.
The definition of soil health is close to the soil quality concept introduced in the 1990's:"*the capacity of the soil to function*
*within ecosystem and land-use boundaries to sustain productivity, maintain environmental quality and promote plant and*
*animal health"* (Bouma, 2002; Bünemann et al., 2018; Doran and Parkin, 1994; Karlen et al., 1997). Discussions in the
early 2000's have resulted in a distinction between *inherent* and *dynamic* soil quality. The former would be based on
relatively stable soil properties as expressed in soil types that reflect the long-term effect of the soil forming factors
corresponding with the basic and justified assumption of soil classification that soil management should not change a given



classification. Still, soil functioning of a given soil type can vary significantly as a result of the effects of past and current
soil management, even though the name of the soil type does not change (this can be the soil series as defined in USDA
Soil Taxonomy (Soil Survey Staff, 2014 as expressed in Table 1). *Dynamic* soil quality would reflect possible changes as
a result of soil use and management over a human time scale, which can have a semi-permanent character when considering
, for example, subsoil plowpans (e.g. Mobius-Clune et al, 2016). This was also recognized by Droogers and Bouma, (1997)
and Rossiter and Bouma (2018) when defining different soil phenoforms reflecting effects of land use for a given genoform
as distinguished in soil classification. Distinction of different soil phenoforms was next translated into a range of
characteristic different soil qualities by using simulation techniques (Bouma and Droogers, 1998). Soil health at a given
time could next be considered to represent actual quality conditions, fitting into this particular soil quality range.
The term soil health appears to have a higher appeal for land users and citizens at large than the more academic term soil
quality, possibly because the term "health" has a direct connotation with human wellbeing in contrast to the more distant
and abstract term: "quality". Humans differ and so do soils; some soils are genetically more healthy than others and a given
soil can have different degrees of health at any given time, which depends not only on soil properties but also on past and
current management and weather conditions. Mobius-Clune et al, 2016 have recognized the importance of climate variation
by stating that their proposed system only applies to the North-East of the USA and its particular climate and soil conditions.
This represents a clear limitation and could in time lead to a wide variety of local systems with different parameters that
would inhibit effective communication to the outside world. This paper will therefore explore possiblities for a systems
approach with general applicability. To apply the soil health concept to a wider range of soils in other parts of the world,
the attractive analogy with human health not only implies that "health" has to be associated with particular soil individuals
( usually expressed in terms of a given soil series), but also to climate zones. In addition, current questions about soil
behavior often deal with possible effects of climate change. In this paper, the proposed systems analysis can – in contrast
to the procedures presented so far- also deal with this issue. Using soils as a basis for the analysis is only realistic when soil
types can be unambiguously defined, as was demonstrated by Bonfante and Bouma (2015) for five soil series in the Italian
Destre Sele area. In most developed countries where soil surveys have been completed, soil databases provide extensive
information on the various soil series, including parameters needed to define soil quality and soil health in a systems-
analysis as shown, for example, for clay soils in the Netherlands (Bouma and Wösten, 2016). The recent report of the
National Academy of Sciences, Engineering and Medicine (2018) also emphasizes the need for a systems approach.
The basic premise of the Soil Health concept, as advocated by Moebius-Clune et.al. 2016 and others, is convincing. Soil
characterization programs since the early part of the last century have been exclusively focused on soil chemistry and soil
chemical fertility and this has resulted in not only effective recommendations for the application of chemical fertilizers but
also in successful pedological soil characterization research. But soils are living bodies in a landscape context and not only
chemical but also physical and biological processes govern soil functions. The Soil Health concept considers therefore not
only soil chemical characteristics, that largely correspond with the ones already present in existing soil fertility protocols,
but also with physical and biological characteristics that are determined with well defined methods, with particular emphasis
on soil biological parameters (Moebius-Clune et al, 2016). However, the proposed soil physical methods by Moebius-Clune
et al ( 2016) don't reflect modern soil physical expertise and procedures need to have a universal rather than a regional
character, while pressing questions about the effects of soil degradation and future climate change need to be addressed as
well. The proposed procedures do not allow this. Explorative simulation studies can be used to express possible effects of
climate change as, obviously, measurements in future are not feasible. Also, only simulation models can provide a



quantitative, interdisciplinary integration of soil-water-plant-atmosphere processes that are key to both the soil quality and
soil health definitions, as mentioned above.
In summary, the objectives of this paper are to: (i) explore alternative procedures to characterize: "soil physical quality
and health" applying a systems analysis by modeling the soil-water-plant-atmosphere system, an analysis that is valid
anywhere on earth ; (ii) apply the procedure to develop quantitative expressions for the effects of different forms of soil
degradation, and (iii) explore effects of climate change for different soils also considering different forms of soil
degradation. Expressions for chemical and biological soil health will not be discussed here but are needed to be integrated
with the soil physical analysis, to allow a classification of overall soil health.

## 116 2. Materials and methods

### 117 2.1. Soil functions as a starting point

The soil quality and - health definitions both mention: "*the continued capacity of a soil to function*". Soil functions have
therefore a central role in the quality and health debate. EC (2006) defined the following soil functions: (1) Biomass
production, including agriculture and forestry; (2) Storing, filtering and transforming nutrients,substances and water: (3)
Biodiversity pool, such as habitats, species and genes; (4) Physical and cultural environment for humans and human
activities; (5) Source of raw material; (6) Acting as carbon pool, and (7) Archive of geological and archaeological heritage.
Functions iv, v and vii are not covered in this contribution since they are considered special as they require, if considered
relevant, specific measures to set soils apart by legislative measures. The other functions are directly and indirectly related
to function 1, biomass production. Of course, soil processes not only offer contributions to biomass production, but also to
filtering, biodiversity preservation and carbon storage. Inter- and transdisciplinary approaches are needed to obtain a
complete characterization, requiring interaction with other disciplines, such as agronomy, hydrology, ecology and
climatology and, last but not least, with stakeholders and policy makers. Soil functions thus contribute to ecosystem services
and, ultimately, to all seventeen UN Sustainable Development Goals (e.g. Bouma, 2016, 2014; Keesstra et al., 2016).
However, in the context of this paper, attention will be focused on the soil functions.
Soil physical aspects play a crucial role when considering the role of soil in biomass production, as expressed by Function
1, which is governed by the dynamics of the soil-water-plant-climate system: (1) Roots provide the link between soil and
plant. Rooting patterns as a function of time are key factors for crop uptake of water and nutrients. Deep rooting patterns
imply less susceptibility to moisture stress. Soil structure, the associated bulk densities, and the soil water content determine
whether or not roots can penetrate the soil. When water contents are too high, either because of the presence of a water
table or of a dense, slowly permeable soil horizon impeding vertical flow, roots will not grow because of lack of oxygen.
For example, compact plow-pans, resulting from applying pressure on wet soil by agricultural machinery, can strongly
reduce rooting depth. In fact, soil compaction is a major form of soil degradation that may affect up to 30% of soils in some
areas. (e.g. FAO & ITPS, 2015).
(2) Availability of water during the growing season is another important factor that requires, for a start, infiltration of all
rainwater into the soil and its containment in the unsaturated zone, constituting "green-water" (e.g. Falkenmark and
Rockström, 2006). When precipitation rates are higher than the infiltrative capacity of soils water will flow laterally away
over the soil surface, possibly leading to erosion and reducing the amount of water available for plant growth, and:





(3) the climate and varying weather conditions among the years govern biomass production. Rainfall varies in terms of
quantities, intensities and patterns. Radiation and temperature regimes vary as well. In this context, definitions of location-
specific potential yield (Yp), water-limited yield (Yw) and actual yield (Ya) are important, as will be discussed later .
Soil Function 2 requires soil infiltration of water in the first place followed by good contact between percolating water and
the soil matrix, where clay minerals and organic matter can adsorb cations and organic compounds, involving chemical
processes that will be considered when defining soil chemical quality. However, not only the adsorptive character of the
soil is important but also the flow rate of applied water that can be affected by climatic conditions or by management when
irrigating. Rapid flow rates generally result in poor filtration as was demonstrated for viruses and fecal bacteria in sands
and silt loam soils (Bouma, 1979).
Soil Functions 3 and 6 are a function of the organic matter content of the soil the quantity of which is routinely measured
in chemical soil characterization programs (also in the soil health protocols mentioned earlier that also define methods to
measure soil respiration). The organic matter content of soils is highly affected by soil moisture regimes and soil chemical
conditions. Optimal conditions for rootgrowth in terms of water, air and temperature regimes will also be favorable for soil
biological organisms, linking soil functions 1, 3 and 6.
When defining soil physical aspects of soil quality and soil health, focused on soil function 1, parameters will have to be
defined that integrate various aspects, such as: (1) weather data, (2) the infiltrative capacity of the soil surface, considering
rainfall intensities and quantities, (3) rootability as a function of soil structure, defining thresholds beyond which rooting is
not possible, and: (4) hydraulic and root extraction parameters that allow a dynamic characterization of the soil-water-plant-
atmosphere system that can only be realized by process modeling, that requires these five parameters and modeling is
therefore an ideal vehicle to realize interdisciplinary cooperation.
**2.2. The role of dynamic modeling of the soil-water-plant-atmosphere system**
When analysing soil quality and soil health, emphasis must be on the dynamics of *vital, living ecosystems* requiring a
dynamic approach that is difficult to characterize with static soil characteristics (such as bulk density, organic matter content
and texture) except when these characteristics are used as input data into dynamic simulation models of the soil-water-
plant-climate system. Restricting attention to soil physical characteristics, hydraulic conductivity (K) and moisture retention
properties (h-theta) of soils are applied in such dynamic models.Measurement procedures are complex and can only be
made by specialists, making them unsuitable for general application in the context of soil quality and health. They can,
however, be easily derived from *pedotransferfunctions* that relate static soil characteristics such bulk density, texture and
%C to these two properties, as recently summarized by Van Looy et al., (2017). The latter soil characteristics are available
in existing soil databases and are required information for the dynamic models characterizing the soil production function.
Simulation models of the soil-water-plant-atmosphere system, such as the Soil Water Atmosphere, Plant model (SWAP)
(Kroes et al., 2008) to be discussed later in more detail, integrate weather conditions, infiltration rates, rooting patterns and
soil hydrological conditions in a dynamic systems approach that also allows exploration of future conditions following
climate change. The worldwide agronomic Yield-Gap program (www.yieldgap.org) can be quite helpful when formulating
a soil quality and – health program with a global significance. So-called water-limited yields (Yw) can be calculated,
assuming optimal soil fertility and lack of pests and diseases (e.g Gobbett et al., 2017; van Ittersum et al., 2013; Van Oort
et al., 2017). Yw reflects climate conditions at any given location in the world as it is derived from potential production
(Yp) that reflects radiation, temperature and basic plant properties, assuming that water and nutrients are available and pests



and diseases don't occur. Yw reflects local availability of water and is always lower than Yp. Yw can therefore act as a
proxy value for physical soil quality, focusing on function 1.
Actual yields (Ya) are often, again, lower than Yw (e.g. Van Ittersum et al, 2013). The ratio Ya/Yw is an indicator of the
so-called "yield-gap" showing how much potential there is at a given site to improve production ([www.yieldgap.org](http://www.yieldgap.org))
(Bouma, 2002). When multiplied with 100, a number between 1 and 100 is obtained as a quantitative measure of the "yield
gap" for a given type of soil . Yw can be calculated for a non-degraded soil. Ya shlould ideally be measured but can also
be calculated in this exploratory study (in terms of Yw) on the basis of the assumed effects of different forms of soil
degradation, such as subsoil soil compaction, poor water infiltration at the soil surface due to surface compaction or crusting
and erosion. This requires introduction of a compact layer (plowpan) in the soil,  a reduction of  rainfall amounts with the
volume of overland flow and by removing topsoil. This was done in this exploratory study but, ideally, field observations
should be made in a given soil type to define effects of management as explored by.Pulleman et al., (2000) for clay soils
and  Sonneveld et al., (2002) for sandy soils. Such field work also includes emphasis on important interaction with farmers
as mentioned by Moebius-Clune et al, (2016). Sometimes, soil degradation processes, such as erosion, may be so severe
that the soil classification (the soil genoform) changes. Then, the soil quality and soil health discussion shifts to a different
soil type.
This approach will now be explored with a particular focus on the Mediterranean environment. Physical soil quality is
defined by Yw for each soil, considering a soil without assumed degradation phenomena (the reference) and for three
variants (hypothetical Ya, expressed in terms of Yw) with: (1) a compacted plowlayer, (2) a compacted soil surface resulting
in overland flow, and (3) removal of topsoil following erosion, without a resulting change in the soil classification. This
way a characteristic range of Yw values is obtained for each of the six soil series, reflecting positive and negative effects
of soil management and representing a range of soil quality values of the particular soil series considered. Within this range
an actual value of Ya will indicate the soil physical health of the particular soil at a given time and its position within the
range of values will indicate the severity of the problem and potential for possible improvement.
The ratio (Ya/Yw)x100 is calculated to obtain a numerical value that represents "soil health" as a point value, representing
actual conditions. Health is relatively low when real conditions occur in the lower part of the soil quality range for that
particular soil and relatively high when it occurs in the upper range. Again, in this exploratory study measured values (at
current climate conditions) for Ya have not been made, so Ya only applies to the three degraded soil forms being
distinguished here where hypothetical effects of soil degradation have been simulated as related to the corresponding
calculated Yw values. Of course, actual measured Ya values can't be determined at all when considering future climate
scenario's.
To allow this, attention will be paid to the possible effects of climate change applying RCP 8.5- IPCC scenario. Obviously,
only computer simulations can be used when exploring future conditions, another important reason to use dynamic
simulation modeling in the context of characterizing soil quality and soil health. The approach in this paper extends earlier
studies on soil quality for some major soil types in the world that did not consider aspects of soil health nor effects of
climate change (Bouma, 2002; Bouma et al., 1998).

**2.3. Simulation modeling**
The Soil–Water–Atmosphere–Plant (SWAP) model (Kroes et al., 2008) was applied to solve the soil water balance. SWAP
is an integrated physically-based simulation model of water, solute and heat transport in the saturated–unsaturated zone in




relation to crop growth. In this study only the water flow module was used; it assumes unidimensional vertical flow
processes and calculates the soil water flow through the Richards equation. Soil water retention θ(h) and hydraulic
conductivity K(θ) relationships as proposed by van Genuchten (1980) were applied. The unit gradient was set as the
condition at the bottom boundary. The upper boundary conditions of SWAP in agricultural crops are generally described
by the potential evapotranspiration $ET_p$, irrigation and daily precipitation. Potential evapotranspiration was then partitioned
into potential evaporation and potential transpiration according to the LAI evolution, following the approach of Ritchie
(1972). The water uptake and actual transpiration were modeled according to Feddes *et al.* (1978), where the actual
transpiration declines from its potential value through the parameter α, varying between 0 and 1 according to the soil water
potential.
The model was calibrated and validated by measured soil water content data at different depths for Italian conditions
(Bonfante et al., 2010; Crescimanno and Garofalo, 2005) and in the same study area by (Bonfante et al., 2011, 2017). In
particular, the model was evaluated in two farms inside of Destra Sele area, on three different soils (Udic Calciustert,
Fluventic Haplustept and Typic Calciustoll), under maize crop (two cropping seasons) during a Regional project "Campania
Nitrati" (Regione Campania, 2008) (Tab.2).
Soil hydraulic properties of soil horizons in the area were estimated by the pedotransfer function (PTF) HYPRES (Wösten
et al., 1999). A test of reliability of this PTF was performed on θ(h) and k(θ) measured in the laboratory by the evaporation
method (Basile et al., 2006) on 10 undisturbed soil samples collected in the Destra Sele area. The data obtained were
compared with estimates by HYPRES and were considered to be acceptable (RMSE = 0.02 $m^3$ $m^{-3}$) (Bonfante et al., 2015).
Simulations were run considering a soil without assumed degradation phenomena (the reference) and for three variants with
a compacted plowlayer, surface runoff and erosion, as discussed above:
(i) The compacted plowlayer was applied at -30cm (10 cm of thickness) with following physical characteristics: 0.30 WC
at saturation, 1.12 n, 0.004 "a" and Ks of 2 cm/day. Roots were restricted to the upper 30 cm of the soil. (ii) Runoff from
the soil surface was simulated removing ponded water resulting form intensive rainfall events. Rooting depth was assumed
to be 80 cm. (iii) Erosion was simulated for the Ap horizon, reducing the upper soil layer to 20 cm. The maximum rooting
depth was assumed to be 60 cm (A+B horizon) with a higher root density in the Ap horizon.
Variants were theorical but based on local knowledge of the Sele Plain. Compaction is relevant considering the highly
specialized and intensive horticulture land use of the Sele plain which typically involves repetitive soil tillage at similar
depth.  Runoff and erosion easily occur at higher altitude plain areas especially where the LON0, CIF0/RAG0, GIU0 soil
types occur (Fig. 1).

**2.4. Soils in the Destra Sele area in Italy.**
The "Destra Sele" study area, the plain of the River Sele (22,000 ha, of which 18,500 ha is farmed) is situated in the south
of Campania, southern Italy (Fig. 1). The main agricultural production consists of irrigated crops (maize, vegetables and
fruit orchards), greenhouse-grown vegetables and mozzarella cheese from water buffalo herds. The area can be divided into
four different environmental systems (hills/footslopes, alluvial fans, fluvial terraces and dunes) with heterogeneous parent
materials in which twenty different soil series were distinguished (within Inceptisol, Alfisol, Mollisol, Entisol and Vertisol
soil orders) (Regione Campania, 1996), according to Soil Taxonomy (Soil, 1999). Six soil series were selected in the area
to test application of the soil quality and soil health concepts. Representative data for the soils are presented in Table 1.
Decision trees were developed to test whether the selection process of the soil series was based on stable criteria, allowing
extrapolation of results from measured to unmeasured locations when considering effects of climate change. While





extrapolation in space of soil series data has been a common procedure in soil survey (e.g. Soil Survey Staff, 2014; Bouma
et al., 2012), extrapolation in time has not received as much attention. A basic principle of many taxonomic soil
classification systems is a focus on stable soil characteristics when selecting diagnostic criteria for soil types. Also, emphasis
on morphological features allows, in principle, a soil classification without requiring elaborate laboratory analyses. (e.g.
Soil Survey Staff, 2014). A given soil classification should not change following plowing or other management measures
as long as this does, of course, not result in removal of soil or in invasive anthropic activity. This way, soil classification
results in an assessment of the (semi)-permanent physical constitution of a given soil in terms of its horizons and textures.
That is why soil quality is defined for each soil type as a characteristic range of Yw values, representing different effects
of soil management that have not changed the soil classification.
**2.5. Climate information**
Future climate scenario were obtained by using the high resolution regional climate model (RCM) COSMO-CLM (Rockel
et al., 2008), with a configuration employing a spatial resolution of 0.0715°(about 8 km), which was optimised over the
Italian area. The validations performed showed that these model data agree closely with different regional high-resolution
observational datasets, in terms of both average temperature and precipitation in Bucchignani et al. (2015) and in terms of
extreme events in Zollo et al. (2015).
In particular, the RCP[1] 8.5 scenario was applied, based on the IPCC (Intergovernmental Panel on Climate Change)
modelling approach to generate greenhouse gas (GHG) concentrations (Meinshausen et al., 2011). Initial and boundary
conditions for running RCM simulations with COSMO-CLM were provided by the general circulation model CMCC-CM
(Scoccimarro et al., 2011), whose atmospheric component (ECHAM5) has a horizontal resolution of about 85 km. The
simulation was performed cover the period from 1979 to 2100; more specifically, the CMIP5 historical experiment (based
on historical greenhouse gas concentrations) was used for the period 1976–2005 (Reference Climate scenario - RC), while,
for the period 2006–2100, a simulation was performed using the IPCC scenario mentioned.
Daily reference evapotranspiration (ET$_0$) was evaluated according to Hargreaves and Samani, (1985) equation (HS). The
reliability of this equation in the study area was perrformed by Fagnano et al., (2001) comparing the HS equation with the
Penman–Monteith (PM) equation (Allen et al., 1998).

**3.   RESULTS AND DISCUSSION**
**3.1. Soil physical quality of the soil series, as expressed by Yw, under current and future climates.**
Soil physical quality of the six soil series, expressed as calculated Yw values for the
reference climate and for future climate scenario RCP 8.5, expressed for three time windows are
shown in Figure 2. Considering current climate conditions, the Longobardo and Cifariello soils with
loamy textures have the highest values, while the sandy soil Lazzaretto is lower. This can be explained
by higher water retention of loamy soils (180 and 152 mm of AWC in the first 80 cm for Longobarda
and Cifariello respectively) compared to the sandy soil (53 mm of AWC in the first 80 cm for
Lazzaretto). The effects of climate change are most pronounced and quite clear for the two
periods after 2040. Reductions compared with the period up to 2040 range from 20-40%, the highest
values associated with sandier soil textures. This follows from the important reduction of projected rainfall during the cropping
season (Fig. 3) ranging from an average value of 235 (±30) mm in the 2010-2040 period to 185 (±26) mm (-21%) and to 142

---

[1] Representative Concentration Pathway





(±24) mm (-40%) in the 2040-2070 and 2070-2100 periods, respectively (significant at p< 0.01). The figure also includes a
value for Yp, potential production (under RC with optimal irrigation), which is 18 t ha$^{-1}$, well above the Yw values.
Only a Yp value is presented for current conditions because estimates for future climates involve too
many unknown factors.

**3.2. Projected effects of soil degradation processes**
*3.2.1. Projected effects of subsoil compaction.*
The projected effects of soil compaction are shown in Figure 4. The effects of compaction are very
strong in all soils, demonstrating that restricting the rooting depth has major effects on soil production.
Compared with the reference, reductions in Yw do not occur in the first time window (2010-2040), as a function of the soil
characteristics of the upper 30 cm of the soils, while the projected lower precipitation rates in future climates will have a
significant effect on all soils, strongly reducing Yw values by 44-55% with, again, highest values in the sandy soils. Clearly,
any effort to increase effective rooting patterns of crops should be a key element when considering attampts to combat effects
of climate change. Data indicate that reactions are soil specific.
*3.2.2. Projected effects of overland flow.*
Results, presented in Figure 5, show relatively small differences (5% or less) with results presented in Figure 2 that was based
on complete infiltration of rainwater. This implies that surface crusting or compaction of surface soil, leading to lower
infiltration rates and more surface runoff, does not seem to have played a major role here in the assumed scenario's. Real field
measurements may well produce different results. Even though projected future climate scenario's predict rains with higher
intensities, that were reflected in the climate scenario's being run, the effects of lower precipitation, as shown in Figure 3,
appear to dominate.
*3.2.3. Projected effects of erosion.*
Results, presented in Figure 6, show significant differences with results presented in Figure 2. Yw values are lower in all soils
as compared with reference climate conditions, but loamy and clayey subsoils still can still provide moisture to plant roots,
leading to relatively low reductions of Yw (e.g 10%-20% for the Longobarda and Cifariello soils, with an AWC to the
remaining 60 cm depth of 150 mm and 120 mm, respectively) even though topsoils with a relatively high organic matter
content have been removed.  Next are the Picciola, Giuliarossa and San Vito soils with reductions between 35 and 45%, all
with an AWC of appr. 107 mm. Effects of erosion are strongest in the sandy Lazzaretto soil, where loss of the A horizon has
a relatively strong effect on the moisture supply capacity of the remaining soil with an AWC of 33 mm up to the new 60 cm
depth. The reduction with the reference level is 30%, which is relatively low because the reference level was already low as
well. Projected effects of climate change are, again, strong for all soils, leading to additional reductions of Yw of appr. 30%.
*3.2.4. Indicators for the soil quality range.*
Figure 7 presents the physical soil quality ranges for the six soils, expressed separately as bars for each of the four climate
periods. The (Ya/Yw) x100 index illustrates that ranges are significantly different. The upper limit is theoretically 100%. But
Van Ittersum et al (2013) have suggested that an 80% limit would perhaps be more realistic and this limit is indicated in Figure
7, where the lower limits for the range vary from e.g. 35 (Longobarda) to 55 (Lazzaretto) for the reference climate with other
values in between and decrease as the projected reaction to climate change (e.g. 20 for Longobardo and 40 for Lazzaretto). This
provides important signals for the future.
As discussed, the presented ranges are soil specific and are based on hypothetical conditions associated with different forms
of land degradation. Field research may well result in different ranges also possibly considering different soil degradation



factors beyond compaction, surface runoff and erosion. Still, principles involved are identical. Ranges presented in Figure 7
represent a physical soil quality range that is characteristic for that particular type of soil. Actual values (Ya) will fit somewhere
in this range and will thus indicate how far they are removed from the maximum and minimum value, presenting a quantitative
measure for soil physical health. This can not only be important for communication purposes but it also allows a judgment of
the effects of different forms of degradation in different soils as well as potential for improvement.

**4.  Discussion**
Linking the soil quality and soil health discussion with the international research program on the *yield gap* allows direct and
well researched expressions for crop yields, defining soil function 1, as discussed above. The potential yield (Yp) and water-
limited yield (Yw) concepts apply worldwide and provide therefore, a sound theoretical basis for a general soil quality/health
classification, avoiding many local and highly diverse activities as reviewed by Büneman et al, (2017). Of course, different
indicator crops will have to be defined for different areas in the world.
Linking soil quality and health to specific and well defined soil types is essential because soil types, such as the soil series
presented in this paper, uniquely reflect soil forming processes in a landscape context. They provide much more information
than just a collection of soil characteristics, such as texture, organic matter content and bulk density. They are well known to
stakeholders and policy makers in many countries. A good example is the USA where State Soils have been defined.
Defining (semi-permanent) soil quality for specific soil types, in terms of a characteristic range of Yw values reflecting
effects of different forms of land management, represents a quantification of the more traditional Soil Survey interpretations
or land evaluations where soil performance was judged by qualitative, empirical criteria. (e.g. FAO, 2007, Bouma et al

358  2012).

In this exploratory study, hypothetical effects of three forms of soil degradation were tested.  In reality, soil researchers
should go to the field and assemble data for a given soil series as shown on soil maps, establishing a characteristic range of
properties, following the example of Pulleman et al (2000) for a clay soil and Sonneveld et al, (2002) for a sand soil, but not
restricting attention to %C but including al least bulk density measurements. This way, soil quality (based on the genoform)
has a characteristic range of Yw values, as shown in Figure 7. Soil physical health at any given time is reflected by the
position of real Ya values within that range and can be expressed by a number (Ya/Yw) x100.
One could argue that this "range" acts as a "thermometer" for a particular type of soil allowing determination of the physical
"health" of a given soil by the placement of Ya. But calculating Yw has implications beyond defining physical soil quality
and health. It can function as a starting point of the general soil quality/soil health discussion. As discussed, Yw assumes that
nutrients, pests and diseases don't inhibit biomass production. If Ya is lower than 80% of Yw the reasons must be found.
Chemical conditions in the soil that affect plant growth may be a reason, as may be unfavorable biological conditions or poor
soil management. Tillage practices, crop rotations or poor handling pests and diseases may be reasons as well. This will cover
soil functions 2, 3 and 6, as discussed above completing consideration of all soil functions.
Rather than consider the physical, chemical and biological aspects separately, each with their own Indicators as proposed by
Moebius-Clune et al, (2016), following a logical and interconnected sequence considering pedological, physical, chemical and
iological aspects could be more effective. This is the more relevant because definition of reproducible biological soil health
parameters are still object of study (Wade et al., 2018) and %C might be an acceptable proxy for the time being. Recent tests
of current soil-health protocols have not resulted in adequately expressing soil conditions in North Caolina (Roper et al, 2017),
indicating the need for further research as suggested in this paper.



**5. Conclusions**

1. Lack of widely accepted, operational criteria to express soil quality and soil health is a barrier for effective external communication of the importance of soil science

2. Using well defined soil types as "carriers" of information on soil quality and soil health can improve communication to stakeholders and the policy arena.

3. A universal system defining soil quality and soil health is needed based on reproducible scientific principles that can be applied all over the world, avoiding a multitude of different local systems. Models of the soil-water-plant-atmosphere system can fulfil this role.

4. Connecting with the international *yield gap* program, applying soil-water-plant-atmosphere simulation models, will facilitate cooperation with agronomists which is essential to quantify the important soil function 1: biomass production.

5. Cooperation and initiating a joint-learning process with stakeholders and policy makers is essential to achieve acceptance of derived protocols.

6. The proposed system allows an extension of classical soil classification schemes, defining genoforms, by allowing estimates of effects of various forms of past and present soil management (phenoforms) within a given genoform that often strongly affects soil performance. Quantitative information thus obtained can improve current empirical and qualitative soil survey interpretations and land evaluation.

7. Rather than consider physical, chemical and biological aspects of soil quality and - health separately, a combined approach starting with pedological and soil physical aspects followed by chemical and biological aspects, all to be manipulated by management, is to be preferred.

8. Only the proposed modeling approach allows exploration of possible effects of climate change on future soil behaviour which is a necessity considering societal concerns and questions.

9. Field work, based on existing soil maps to select sampling locations for a given genoform, is needed to identify a characteristic range of phenoforms for a given genoform, which, in turn, can define a characteristic soil quality range by calculating Yw values.

**6  Acknowledgements**

We acknowledge Dr. Eugenia Monaco and Dr. Langella Giuliano for the supporting in the analysis of climate scenario. The "Regional Models and Geo-Hydrogeological Impacts Division", Centro Euro-Mediterraneo sui Cambiamenti Climatici (CMCC), Capua (CE) – Italy, and the Dr. Paola Mercogliano and Edoardo Bucchignani for the future climate scenario applied in this work.



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




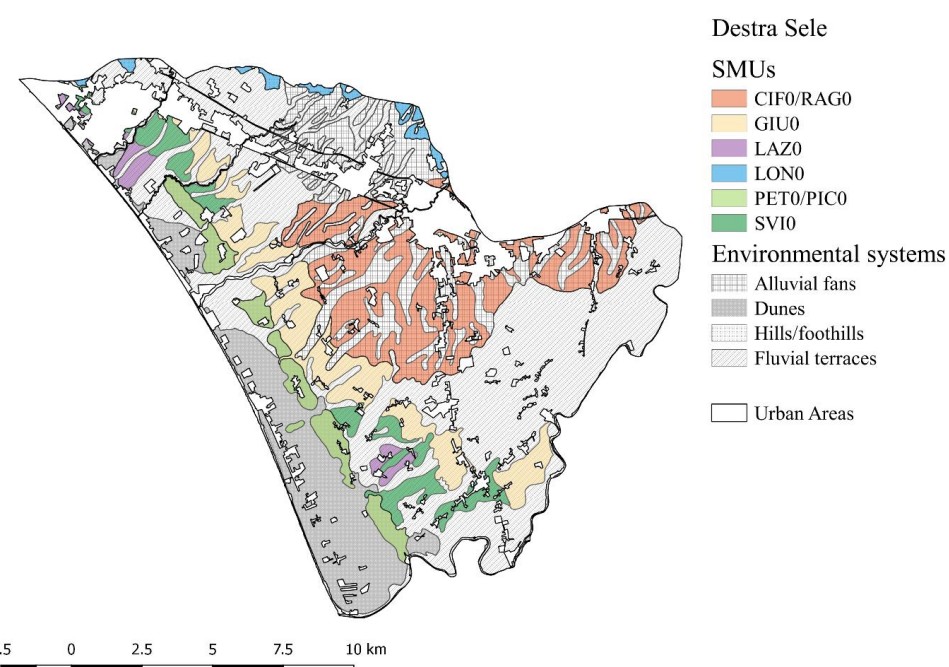


**Figure 1: The four environmental systems of the "Destra Sele" area and the Soil Map Units (SMU) of selected six Soil Typological Units (STUs, which are similar to the USDA soil series) (CIF0/RAG0= Cifariello; GIU0= Giuliarossa; LAZ0= Lazzaretto; LON0= Longobarda; PET0/PIC0= Picciola; SVI= San Vito).**




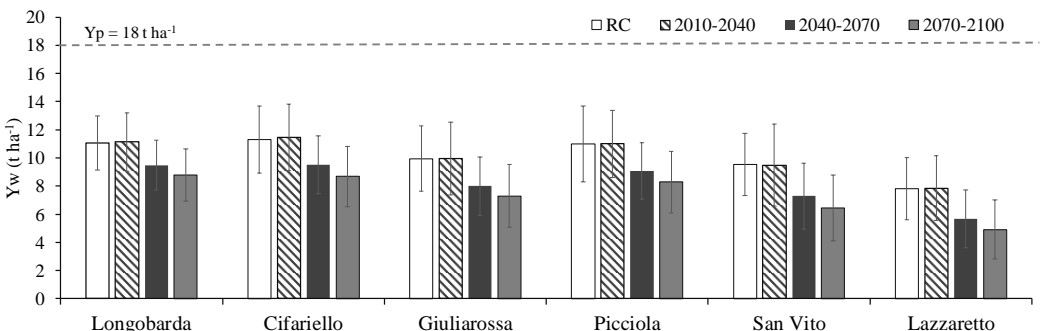

**Figure 2: Simulated Yw values for six soil series, considering the reference climate (1976-2005) and future climate scenario's RCP 8.5 expressed in three time windows (2010-2040; 2040-2070; 2070-2100). The Yp (potential yield) is the average production with optimal irrigation under reference climate calculated over all soil series.**

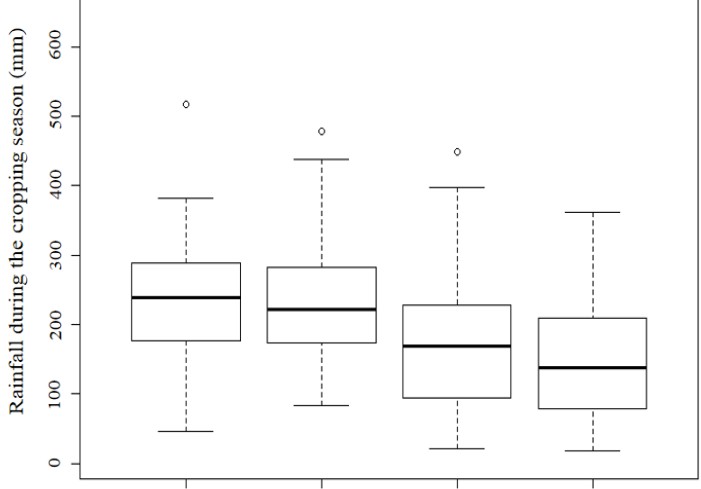

**Figure 3: Cumulated rainfall during the maize growing season (April–August) in the four climate periods.**




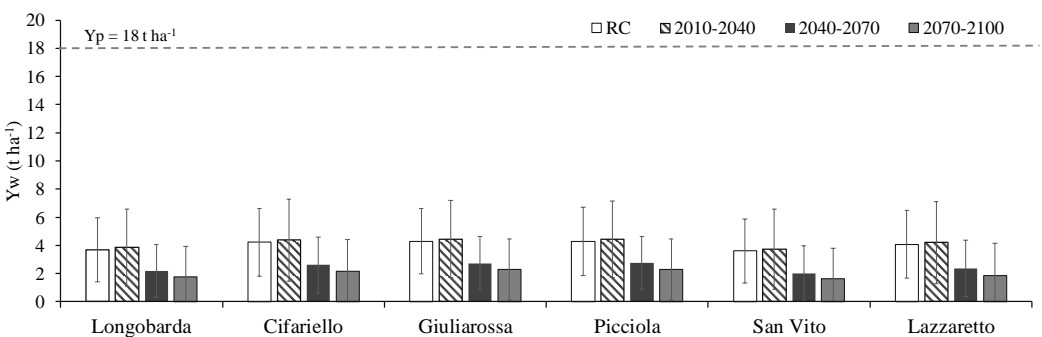

**Fig.4 The projected effects of simulated soil compaction on Yw in the four climate periods, assuming the presence of a compacted plowlayer at 30 cm depth. The Yp (potential yield) is the average production with optimal irrigation, under reference climate calculated over all soil series under reference soil conditions.**

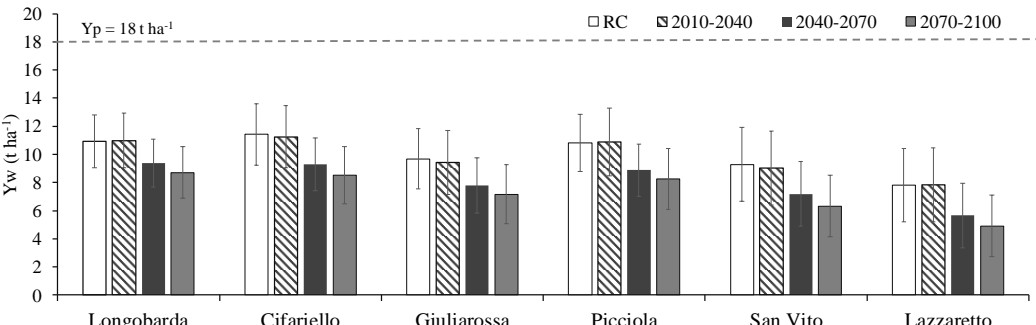

**Figure 5: The projected effects of simulated surface runoff of water on Yw in the four climate periods, occurring when precipitation rates exceed the infiltrativce capacity of the soil.**




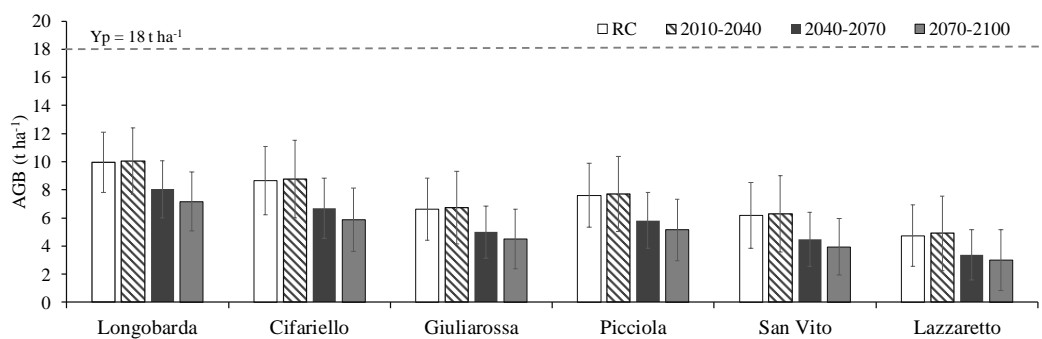

**Figure 6: The projected effects of simulated Yw following erosion, reducing to 20 cm the topsoil. Results are reported for the four climate periods.**


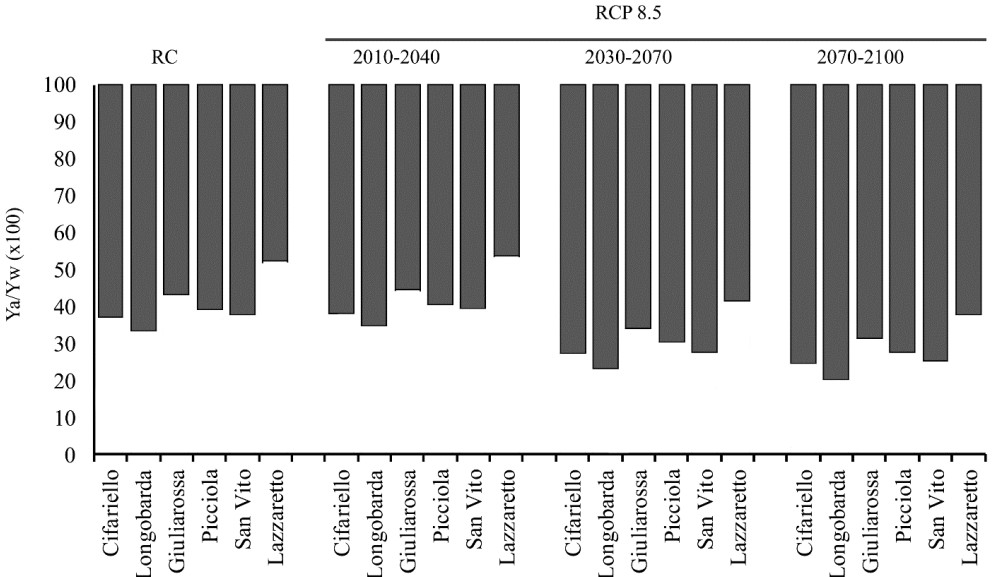

**Figure 7: Range of soil physical quality indexes (Ya/Yw) x 100) for the six soils, expressed for four different climate periods.**

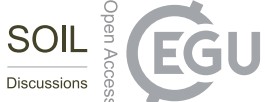


Tab. 1. Main soil features of selected soil series.

| Env. Systems | SMU | STU | Soil family | Soil description | | Texture | | | Hydrological properties | | | | |
|---|---|---|---|---|---|---|---|---|---|---|---|---|---|
| | | | | Horiz. | Depth (m) | sand | silty | clay | $\Theta s$ | $K_0$ | $\alpha$ | l | n |
| | | | | | | | (g 100g$^{-1}$) | | (m$^3$ m$^{-3}$) | (cm d$^{-1}$) | (1 cm$^{-1}$) | | |
| Hills/foothills | LON0 | Longobarda | Pachic Haploxerolls, fine loamy, mixed, thermic | Ap | 0-0.5 | 33.0 | 40.6 | 26.4 | 0.46 | 27 | 0.04 | -3.44 | 1.15 |
| | | | | Bw | 0.5-1.5 | 21.7 | 48.9 | 29.4 | 0.61 | 69 | 0.02 | -1.79 | 1.18 |
| Alluvial fans | CIF0/ RAG0 | Cifariello | Typic Haploxerepts, coarse loamy, mixed, thermic | Ap | 0-0.6 | 33.0 | 49.5 | 17.5 | 0.42 | 18 | 0.03 | -2.52 | 1.21 |
| | | | | Bw1 | 0.6-0.95 | 33.2 | 50.2 | 16.6 | 0.47 | 37 | 0.03 | -2.14 | 1.20 |
| | | | | Bw2 | 0.95-1.6 | 29.8 | 52.2 | 18.0 | 0.50 | 49 | 0.03 | -2.02 | 1.20 |
| Fluvial Terraces | GIU0 | Giuliarossa | Mollic Haploxeralf, fine, mixed, thermic | Ap | 0-0.4 | 27.1 | 31.9 | 41.0 | 0.47 | 39 | 0.04 | -3.72 | 1.13 |
| | | | | Bw | 0.4-0.85 | 19.8 | 28.9 | 51.3 | 0.49 | 7 | 0.02 | -1.28 | 1.10 |
| | | | | Bss | 0.85-1.6 | 46.3 | 28.8 | 24.9 | 0.40 | 18 | 0.05 | -2.75 | 1.16 |
| | SVI0 | San Vito | Typic Haploxererts fine, mixed, thermic | Ap | 0-0.5 | 17.3 | 39.4 | 43.3 | 0.44 | 31 | 0.03 | -3.58 | 1.15 |
| | | | | Bw | 0.5-0.9 | 16.1 | 39.6 | 44.3 | 0.49 | 11 | 0.02 | -3.35 | 1.09 |
| | | | | Bk | 0.9-1.3 | 11.2 | 40.5 | 48.3 | 0.49 | 10 | 0.02 | -2.52 | 1.10 |
| | LAZ0 | Lazzaretto | Typic Xeropsamments, mixed, thermic | Ap | 0-0.45 | 75.3 | 12.8 | 11.9 | 0.38 | 77 | 0.07 | -2.26 | 1.30 |
| | | | | C | 0.45- >0.65 | 100.0 | 0.0 | 0.0 | 0.34 | 123 | 0.08 | 2.04 | 1.85 |
| Dunes | PET0/ PIC0 | Picciola | Typic Haploxerepts, coarse loamy, mixed, thermic | Ap | 0-0.6 | 33.3 | 34.7 | 32.0 | 0.48 | 36 | 0.04 | -3.60 | 1.13 |
| | | | | Bw | 0.6-0.95 | 30.5 | 41.2 | 28.3 | 0.44 | 18 | 0.03 | -3.61 | 1.13 |
| | | | | 2Bw | 0.95-1.35 | 28.6 | 50.0 | 21.4 | 0.42 | 21 | 0.03 | -2.77 | 1.17 |





Tab. 2. Main performance indexes of SWAP application in the three soils (Udic Calciustert, Fluventic Haplustept and Typic Calciustoll) under maize cultivation (data from "Nitrati Campania" regional project, Regione Campania, 2008.).

| Soil | RMSE* | R di Pearson* | n° of soil depths meas. | number of data |
|---|---|---|---|---|
| Udic Calciustert | 0.043 (± 0.03) | 0.716 (± 0.11) | 7 | 1964 |
| Typic Calciustoll | 0.044 (± 0.03) | 0.72 (± 0.13) | 6 | 190 |
| Fluventic Haplustept | 0.031(± 0.02) | 0.821 (± 0.09) | 6 | 318 |


*(average value ± standard deviation)*