# Peer review of "Refining physical aspects of soil quality and soil health when"

_SOIL, 2018_

## Referee Comment (RC1) · D. Rossiter (Referee) · 24 Sep 2018

Review of soil-2018-30 D G Rossiter ISRIC-World Soil Information/Cornell University/Nanjing Normal University

(1) General comments

This paper is a welcome step towards quantifying the concept of "soil health" and towards relating it to the concept of soil phenoforms (management-induced semi-permanent changes in soil properties within one soil genoform). It also presents a

convincing argument to use simulation for the future (obviously). The technical aspects are sound, in particular a good choice of soil-plant-atmosphere model and associated pedotransfer functions and a good choice of quantitative phenoform indicators. Less convincing are the future scenarios, although that is entirely because of the uncertainty in the RCP 8.5- IPCC scenario – a reasonable choice since this is what is presented to policy makers. The clear message is that biomass yield, as affected by changes in soil physical properties, can be a quantitative indicator of soil physical "health".

The paper mentions an "logical and interconnected sequence considering pedological, physical, chemical and biological aspects" to holistically evaluate soil health; however the paper does not give any details of how such a sequence would work, nor indeed why a sequential approach (and in the order given, at that) would be desireable. This is outside the scope of the paper (as indicated by its title) but if it is included in the discussion it could be expanded somewhat.

(2) Specific comments

L30 likely under the scenarios; see also comment below on L309

L57 fixed values as expressed by laboratory measurements of the pressure head

L91 Unfortunately, the "soil series" is not used everywhere, explain that the lowest level of other classifications are essentially the same concept. However this level is recognized as necessary for communication with stakeholders, see for example: Lepsch, I. F. (2013). Status of soil surveys and demand for soil series descriptions in Brazil. Soil Horizons, 54(2), 0. https://doi.org/10.2136/sh2013-54-2-gc

L182: Is Yw always lower than Yp? Perhaps if averaged over a number of years – there are always unfavourable years.

L200 These are the phenoforms! emphasize

L255, Figure 1: terminology "environmental systems" seems over-ambitious for what are "landform classes" or similar. Is this the standard terminology used in Italian soil

survey?

L259 It isunclear how these decision trees work

L309 will -> are expected to (under the scenarios); this is correct in o.a. L329 "Projected effects..."

L354 also in the USA soil series names are often used in advertising for farmland, as well as by agricultural consultants

L362, conclusions: Suggest to use the full term "soil [genoforms, phenoforms]" throughout, for consistency with Rossiter & Bouma (2018), cited in the paper.

L373 More detail on what would be a "logical and interconnected sequence considering pedological, physical, chemical and biological aspects", or leave this outside of the scope of this paper.

L385-7 Conclusion point 3. Can these models also cover the biological aspects as proposed in the paper?

L391-2 Conclusion point 5. Not established in this paper.

Figure 2 shows "error bars" for Yw simulations, but neither the text nor the figure caption explain how these are derived. Is this from simulating each year (the uncertainty) and then averaging (the bar)? Similarly for Figure 3, why do we have a boxplot and not just one value per reference? \S2.3 (Simulation modelling) does not make it clear that the simulations were run per year (I think) for each 30-year period. Neither is this clear from \S2.5 (Climate information); L280 only mentions the full period, and does not even break it down into the 30-year intervals that are shown in the Figures and in \S3.1.

Figure 2 caption "over all soil series" is in fact only taking into account the area's climate and an ideal profile, so it has nothing to do with the series.

(3) Technical corrections

Please run a spelling checker

Awkward use of - in abstract

Please be consistent: either Fig. or Figure in the text.

L41 discourse

L55 penetrometer

L62 quote " not "

L69 ) –> –

L95 (2016)

L102-3 run-on sentence with two different ideas

L119-123 and further in this section: inconsistent use of arabic or small roman numerals, with or without parentheses

L132 add: "in three ways": (1)...

L171 pedotransferfunctions -> pedotransfer functions

L211 etc. scenario's [Dutch] -> scenarios [English]

L232 reference to Soil Taxonomy (which version?)

L257 (Soil, 1999) not correct; also L515 in references

L280 "The simulation was performed cover the period" not grammatical; I think this means "The simulations covered the period..."

L372 Indicators -> indicators

L374 [b]iological

L376 Carolina

Figure 5 caption infiltrativce -> infiltration

Table 2 R di Pearson -> Pearson's R
* * *

---

## Author Comment (AC1) · 8 Oct 2018

We thank reviewer D.G.Rossiter for his positive comments on our paper. He considers the paper to be technically sound and a welcome step towards quantifying "soil health"and "soil quality". He supports the use of the phenoform concept to show that a given soil type can act differently as a function of past and current management and he also supports the use of dynamic simulation models to characterize the soil-water-plant-atmosphere system in the context of soil quality and soil health studies. . He raises a question about the :"logical and interconnected sequence considering pedological, physical, chemical and biological aspects". Indeed, this issue does not relate directly to the study on soil physical aspects as presented in this paper. It was added to avoid the impression that soil physical aspects would be a quite seperate entity, next to seperate chemical and biological aspects. This is, in fact, is implicitly suggested in the Cornell procedure where three separate indicators are multiplied. Physical conditions are obviously related to soil moisture regimes in a landscape context, requiring a pedological analysis, and soil chemical conditions are, in turn, related to the physical processes while biological conditions react to both. Physically based models of the soil-water-plant-atmosphere system, as applied in this study, can define conditions that are important for chemical and biological aspects involved in the concept of soil health. For example, the activity of microrganisms involved in the mineralization process of organic matter as well as in the nitrification cycle, is dependent on soil responses (e.g. soil nutrients, moisture status and, soil temperature) and on environmental driving forces (upper and bottom boundaries of the soil-water-plant-atmosphere system). With models it is possible to describe soil behaviour and the resulting enviromental conditions for microorganisms. This allows a distinction between different soils. However, in this paper only the potential for a joint rather than a seperate approach to physical, chemical and biological soil conditions is suggested . As attention in the paper is confined to soil physical conditions, relations with chemical and biological conditions are not further explored but we will explain more clearly in the revised paper (based on text in lines 372-377) the connections between physical, chemical and biological soil processes. This comment also relates to conclusion 3 (lines 385-387). Your specific comments and technical corrections have been noted and the next version of the paper will include your welcome suggestions, for which we are grateful. Some of your comments require a reaction: Line 91: Yes, the soil series concept should be better explained in terms of the lowest level of also other classification systems. Line 182: Yes, when water is always available Yp can be equal to Yw but this is unusual. Line 200: we will emphasize the phenoforms. Line 255: The terminology "environmental sytems" will be changed in "landform classes". L391-2: Correct. Though true, this iss not covered in

the paper and we will omit this conclusion. Figure2 The error bars in Figures 2 and 3 explain the uncertainty, it was derived by the simulation of each year of the specific time period (e.g. 30 years). Moreover, we will specify in line 280 that future climate scenarios have been discussed dividing the whole period in three time windows (2010-2040; 2040-2070 and 2070-2100). The caption will be improved. Figure 3. We decide to report the behaviour of rainfall conditions in terms of variability during the reference period. This is important to show that the reference period and the 2010-2040 period are not so different.

---

## Short Comment (SC1) · 23 Oct 2018

*A note upfront from the submitting person: This review was prepared by Jasmin Kesselring, master students in earth system science at the University of Zurich. The review was part of an exercise during a first semester master level seminar, which I (co-) organize. We would like to highlight that the depth of scientific knowledge and technical understanding of these reviewers represents that of master students. We enjoyed discussing the manuscript in the seminar, and hope that our comments will be helpful for the authors.*

Bonfante et al. use physical soil properties, such as aggregate stability, surface and subsurface hardness and available water capacity, to describe and model soil health and quality. They try to predict the change in theses parameters and therefore in the soil characteristics due to climate change and under different forms of soil degradation. For that, they modelled the soil-water-atmosphere system using a SWAP model and the RPC 8.5- IPCC climate model.

General comments: Bonfante et al. tried to quantify soil health and quality by describing the soil phenoforms rather than soil genoforms. This seems reasonable as the phenoform of the soil takes the past management of soil into account, e.g. possible degradation. To use the IPCC climate model to predict changes in soil health is also quite reasonable as this model is approved by many countries and available globally. However, the results of these climate scenario are somewhat unclear to me. The results of the modelling process are not put into relation with soil health. This could be as the IPCC model has big uncertainties and no clear evolution of the soil in the future can be made. Maybe the influence of future climate on the soil health could be expanded in the discussion.

The paper states clearly, that physical soil properties can be used to quantify soil health. In class we discussed soil health and soil quality and came to the conclusion that it is a function of physical, chemical and biological factors. I do not understand how the soil health can be quantified only using one aspect, when they all influence each other and are dependent on one another. Does the paper conclude that analysing the physical properties is sufficient to derive soil quality? Or were the chemical and biological properties neglected in the paper because they have not been measured?

There are too many conclusions. Not all conclusions are actually discussed in this paper, or are general statements rather than derived from the results of this paper. For example, the conclusions 2 and 5 were not mentioned in the discussion.

The material and methods part of the paper is quite repetitive, and a lot of information
already has been established in the introduction section. For example, the soil physical aspects (L131 pp) have already been mentioned and explained in the introduction section (L44 pp). Further, the chapters 2.2 and 2.3 which are both about the modelling process and mention the same information multiple times. For example, both the SWAP and IPCC climate model are described in 2.3 and 2.2.

– Detailed comments:

Please be consistent when new abbreviations are introduced. Sometimes the explanation with the whole word comes after the first use of the abbreviation.

L90/92 What are these soil series that are described? Is it some form of soil profile archive?

L158-163 This part describes again aspects of the soil function 1. Maybe this could be summed up in one part with the other aspects of function 1 in L131.

L169/222 Be consistent and write either h-theta or $\theta(h)$.

L182 Why does Yw always have to be lower than Yp? Couldn't there be a season with more precipitation than usual.

L210 You state that actual Ya values can't be determined for future scenarios. Would that not be a problem, as you use the Ya/Yw-ratio as the soil health indicator?

L226 What is meant with LAI evolution

L276 Inconsistent use of a footnote

L287 Why is this part called Results and 'Comments' when there is a separate Comments-part later on?

L291/298 Use either Figure or Fig

L385 only the physical properties of the soil where described in this paper. Can the soil-water-plant-atmosphere model also be used for biological or chemical properties?

L400 Why is this approach the only one which allows to explore possible effects of climate change?

Table 2 R di pearson should be Pearson's R

---

## Author Comment (AC2) · 28 Oct 2018

We thank the reviewers dr.Schmidt and his students for their comments. The procedure being followed in preparing this review is innovative and quite interesting. We will tailor our detailed reaction to the composition of this particular review panel. We appreciate the comment that consideration of phenoforms and the IPCC scenario's to express effects of climate change are supported by the reviewers. The first part of their comment suggests that the key message of the paper has apparently not been effectively communicated by us and we will pay particular attention in our revision to

address this issue. As indicated in the paper, both the soil quality and soil health definitions refer to "soil functioning". That's why we focus on defined soil functions, and particularly on function 1: biomass production ( which is also a ecosystem service, not only defined by soil scientists but with input from additional disciplines). That value for biomass production (Ya) can either be measured or estimated by simulation, we have done the latter in this exploratory paper. We are concerned about the statement in the cited recent Cornell bulletin that their procedure is only valid for the NorthWestern USA. The prospect of having many different local systems in future defining soil quality and soil health is not good because lack of a widely accepted general system to define soil quality and health (which we don't have as yet in contrast to water and air quality) forms an increasingly serious barrier to communicate well with citizens, stakeholders and policy makers. We therefore use the worldwide applicable term Yp (potential production), that expresses a yield assuming that water and nutrient supply are optimal and that pests and deseases don't occur. Every soil at any particular location has a characteristic science-based Yp value (for a representative crop at that location). Next we have Yw, which includes the above assumptions, except that it is determined by local water availability and that's why it is usually lower than Yp. When interpreting soil maps, we have to define a "representative"profile for which calculations are made. We introduced the phenoform concept because soil processes are quite different as a function of different forms of management. In our exploratory analysis we assume the presence of a plowpan, surface flow when rainfall exceeds in the infiltrative capacity of the soil and erosion ( assuming that erosion does not change the genoform classification, because our paper is focused on the behavior of individual soil types with a given classification!). In line with Yp, Yw assumes that nutrient supply is optimal and that pests and deseases don't occur. Yw does, therefore, not only address soil physical aspects but also (implicitly) other aspects that affect biomass production. We arbitrarily distinguished three phenoforms, but others can be defined as well, ideally on the basis of field research. Variation of the organic matter content (%C) as a function of soil management is an obvious possibility and provides a link to soil biology as %C

can be seen as a proxy value (we refer to Bouma and Wosten, 2016, and references therein, where a range of %C values is presented for Dutch clay soils) We suggest that soil quality can be represented by a characteristic range of Yw values for any particular type of soil , expressing properties of a series of phenoforms. Actual soil health, based on measured or simulated Ya values, will then have a position within this range. If the distance with Yw is small, health is relatively good and it gets worse as the distance increases. The ratio Ya/Yw x100 provides a soil-specific number which is not only good for communication purposes but also indicates where gains can be achieved (see figure 7). What we have not covered in our paper is what happens next. Your questions refer to that and we will address this in the revised paper. The reason that actual Ya (soil health) has a certain distance to Yw can have many reasons (see the definition of Yw): not only shortage of water but also lack of nutrients or occurrence of pests and diseases. These reasons have to be investigated and corrective measures devised. We focus on Yw and not on Yp because water regimes are relatively difficult to change in contrast to fertility and ocurrence of pests and deseases for which rapid management measures are available (YW can be considered the environmental yield potentiality (soil phenoform + climate) of a specific site). The %C is a proxy for soil biology and different forms of management can increase %C but this may take decades. In this study we have not defined Phenoforms based on different %C of surface soil. In the Cornell protocol, physical, chemical and biological soil quality and health are considered seperately: three numbers are obtained and mutiplied. We suggest that the three processes are highly interrelated and we propose a logical sequence, as mentioned above: start with moisture regimes defining Yw, then analyse why Yw is different from Yp and suggest appropriate management in terms of irrigation or drainage, fertilization practices and application of pest and desease measures. Soil biology can be very dynamic in time and space and it is a function of local hydraulic, physical and chemical soil conditions and %C can act as a proxy for soil quality and health (see soil functions 3 and 6). A proposed management measure can be focused on an increase of %C if this is low in the soil being characterized. . The reviewers present a valuable

comment in that we indeed assume that soil properties don't change as a result of climate change. We will now state so explicitly. Soil forming factors take thousands of years so this seems realistic. We focus on what the effects of compaction etc. may be following climate change. We will remove conclusion 5 . Good point. We have checked the number of repetitions in the Materials and Methods section. We don't believe that sections 2.2.and 2.3 overlap too much. From previous work we have learned that readers are critical about modeling: they want to see all the details. Now we discuss first on modeling in general and next on the details of the model. This seems logical. Detailed comments: Abbreviations will be checked; a reference will be provided with more info on the soil series ( lines l9/92). But Table 1 provides all the necessary data, doesn' t it? Indeed, $Y_w$ can be equal to $Y_p$, but this is unusual. A good point is raised about $Y_a/Y_w$ x100 values in future. Obviously we cannot, in contrast to the present, not measure $Y_a$ values in future. But we can simulate them. In fact this is a major advantage of using simulation of crop growth. (line 210). We have added a sentence. LAI evolution refers to the developemtn of the Leaf Area Index, needed for simulation ( line 226). The soil-water-atmosphere-plant model cannot directly be used to define chemical and biological properties. As explained above, by defining $Y_w$ it can help to focus chemical and biological soil management, based on the gap between $Y_p$ and $Y_w$. And, finally, this study does not necessarily imly that procedures followed are the only way to assess effects of climate change (line 400). Let many flowers bloom!

---

## Referee Comment (RC2) · E. A. Nater (Referee) · 10 Nov 2018

(1) General comments

I very much like the concept of this paper and I think it is a solid step forward in our ability to assess the quality and health of soils worldwide. I especially appreciate the effort to include climate and climate change within a soil health framework. Climate data are critical to development of a universal system of soil health assessment because the spatial and temporal interactions between climate and soil physical properties strongly

influence soil biomass production and other measures of soil health.

The use of dynamic physical and plant growth models based on pedotransfer functions is a powerful tool for assessing soil health over large areas, and is essential for predicting the effects of a changing climate on soil heath and yields.

I strongly agree with the authors that the physical, chemical, and biological aspects of soil health need to be assessed in an interconnected, wholistic fashion, rather than as separate indicators, as there are many potential interactions among them. For example, the biomass yield for soils with moderate salinity may be more strongly impacted by a decrease in precipitation than non-saline soils of similar textures because plant available water would decrease due both to a decrease in precipitation inputs and higher osmotic pressures resulting from a lower dilution of salts present in the solum. I don't believe these types of interactions can be captured by independent Indicators.

(2) Specific comments

L 265: "A given soil classification should not change following plowing or other management measures as long as this does, of course, not result in removal of soil or in invasive anthropic activity." This may or may not be true - the classification of some soils (e.g. shallow Spodosols, thin soils in arid or semi-arid climates) may change following plowing due to complete disruption and mixing of the near surface horizons. The use of deeper tillage instruments has expanded this problem to include soils with thicker sola.

L 295: "The effects of climate change are most pronounced and quite clear for the two periods after 2040." Since most readers are not necessarily familiar with the RCP 8.5 scenario, I believe it would benefit the reader to briefly summarize predicted changes in temperature, precipitation, and possibly PET associated with the RCP 8.5 scenario for this portion of Italy in more detail before describing the results of your own modeling efforts.

L 16-18 and Figs. 4, 5, and 6: The definitions of potential yield (Yp) given in these two locations differs somewhat. The definition provided in the abstract is hypothetical and does not rely on soil properties, whereas the one given in the captions states that it is "... calculated over all soil series under reference soil conditions." These two definitions should be rectified. Fig. 2: Since changes in climate are used to calculate the water-limited yields (Yw) in Figure 2, shouldn't the potential yield (Yp) be adjusted to account for changes in climate? If not, then it should be labeled as Yp (reference climate).

Fig. 3 - It is not clear what the small open circles above the distributions for RC, 2010-2040, and 2040-2070 are. The caption should indicate what these circles represent.

Fig. 7 - this figure is not very clear to me and could be improved.

(3) Technical Corrections

L 187: should for "shlould"

L 243: from instead of form.

L 316 and 317: scenarios rather than "scenario's"

L 370: Tillage practices, crop rotations, or poor handling of pests and diseases...

L 374: biological, not "iological"

Overall this manuscript could be improved by additional English language editing.

---

## Author Comment (AC3) · 15 Nov 2018

We thank Dr.Nater for his very positive evaluation of our paper. His valuable comments lead us to the following reaction and to corresponding modifications and additions in the manuscript. Line 265: The remark on effects of plowing on soil classification is justified. Mentioning of effects of plowing on soil classifications originated from the desire to have permanent names for a given soil that would not change after, for example, plowing. But there are indeed exceptions as the reviewer points out. We have now modified the sentence: A given soil classification should, to obtain permanent names, not change

following traditional management measures, such as plowing. This does, however, not apply to all soils and then a different name will have to be assigned. Line 295. As requested, some additional climatic information is presented for the RCP 8.5 scenario. More information is provided by the cited paper by Bucchignani et al 2015. Line 16-18, and Figs 4,5 and 6. The reviewer is correct: Yp is soil independant. Captions will be improved. We do state that Yp for future climates are unknown, but that can indeed be articulated better by mentioning Yp( reference climate) Figure 3 captions are clarified The additional technical corrections have been made. Thanks.

---

## Author Response (AR1)

[revised manuscript text omitted]

https://doi.org/10.1016/bs.agron.2015.05.001
Bouma, J. 2018. Letter to the Editor. Comment on Minashy and Mc Bratney, 2017. Limited effect of
organic matter on soil available water capacity. Eur.J.of Soil Sci. 69, 154.
(doi:10.1111/ejss.12509).
Bouma, J., 2016. Hydropedology and the societal challenge of realizing the 2015 United Nations
Sustainable Development Goals. Vadose Zo. J. 15.
Bouma, J., 2014.Bouma, J.: Subsurface applications of sewage effluent, Plan. uses Manag. L.,
(planningtheuses), 665–703, 1979.
Bouma, J.: Land quality indicators of sustainable land management across scales, Agric. Ecosyst.
Environ., 88(2), 129–136, doi:10.1016/S0167-8809(01)00248-1, 2002.
Bouma, J.: Soil science contributions towards sustainable development goals and their implementation:
linking soil functions with ecosystem services~.~, J. plant Nutr. soil Sci~.~, 177~,~(2), 111–120, 2014.
Bouma, J., J.J.Stoorvogel and W.M.P.Sonneveld. 2012. Land Evaluation for Landscape Units.
Handbook of Soil Science, Second Edition. P.M.Huang, Y.Li and M..Summer (Eds). Chapter 34.
P.34-1 to 34-22. CRC Press. Boca Raton.London. New York.
Bouma, J.: Hydropedology and the societal challenge of realizing the 2015 United Nations Sustainable
Development Goals, Vadose Zo. J., 15(12), 2016.
Bouma, J., 2002. Land quality indicators of sustainable land management across scales. Agric. Ecosyst.
Environ. 88, 129–136. https://doi.org/10.1016/S0167-8809(01)00248-1
Bouma, J., 1979. Subsurface applications of sewage effluent. Plan. uses Manag. L. 665–703.
Bouma, J., Batjes, N.H., Groot, J.J.R., 1998. Exploring land quality effects on world food supply1.
Geoderma 86, 43–59.
Bouma, J~.~, and Droogers, P~., 1998.~: A procedure to derive land quality indicators for sustainable agricultural production., World Bank Discuss. Pap.., 103–110 [online] Available from: http://www.archive.org/details/plantrelationsfi00coul, 1998.

[revised manuscript text omitted]

National Academy of Sciences, Engineering, Medecine. 2018. Consensus Study Report: Science breakthroughs to advance Food and Agricultural Research by 2030. National Academic Press, Washington DC.

Natural Resources Conservation Services (NRCS): Soil Health. 2012. Retrieved June 23, 2016 from http://www.nrcs.usda.gov/wps/portal/nrcs/main/soils/health/.The Soil Renaissance accepted this definition in 2014

Pulido Moncada, M., Ball, B.C., Gabriels, D., Lobo, D., Cornelis, W.M., 2014a. Evaluation of soil physical quality index S for some tropical and temperate medium-textured soils. Soil Sci. Soc. Am. J. 79, 9–19. http://dx.doi.org/10.2136/sssaj2014.06.0259.

Van Oort, P. A. J., Saito, K., Dieng, I., Grassini, P., Cassman, K. G. and Van Ittersum, M. K.: Can yield gap analysis be used to inform R&D prioritisation?, Glob. Food Sec., 12, 109–118, 2017.

Pulleman, M. M., Bouma, J., Van Essen, E. A., and Meijles, E. W., 2000.: Soil organic matter content as a function of different land use history., Soil Sci. Soc. Am. J., 64, (2), 689–693, 2000.

Regione Campania, 1996.: I Suoli della Piana in Destra Sele. Progetto carta dei Suoli della Regione Campania in scala 1:50.000 e lotto CP1 e Piana destra Sele (Salerno)., 1996.

Regione Campania, 2008. "La ricerca sull'inquinamento da nitrati nei suoli campani: un approccio modellistico nella gestione agro-ambientale". Regione Campania, Assessorato all'Agricoltura ed alle Attività Produttive, SeSIRCA, Napoli 2008. ISBN: 978-88-95230-07-8.

Ritchie, J. T., 1972.: Model for predicting evaporation from a row crop with incomplete cover., Water Resour. Res., 8, (5), 1204–1213, 1972.

Rockel, B., Will, A., and Hense, A., 2008.: The regional climate model COSMO-CLM (CCLM)., Meteorol. Zeitschrift, 17, (4), 347–348, 2008.

Roper, W.H., D.L.Osmond, J.L.Heitman,M.Q.Wagger,S.Ch.Reberg-Horton.2017.Soil health indicators do not differentiate among agronomic managed systems in North Carolina soils. Soil Sci.Soc.Am.J. 81, 828-843. (doi:10.2136/sssaj2016.12.0400)

Rossiter, D. G., and Bouma, J., 2018.: A new look at soil phenoforms--Definition, identification, mapping., Geoderma, 314, 113–121, 2018.

Scoccimarro, E., Gualdi, S., Bellucci, A., Sanna, A., Fogli, P. G., Manzini, E., Vichi, M., Oddo, P., and Navarra, A., 2011.: Effects of Tropical Cyclones on Ocean Heat Transport in a High-Resolution

Coupled General Circulation Model, J. Clim., 24,(16), 4368–4384. https:// doi.org/:Doi 10.1175/2011jcli4104.1, 2011.

Shaw, B. T., Haise, H. R., and Farnsworth, R. B., 1943.: Four Years' Experience with a Soil Penetrometer 1, Soil Sci. Soc. Am. J., 7,(C), 48–55, 1943.

Soil, S.S., 1999. Survey Staff: Keys to soil taxonomy, 1999.

Soil Survey Staff, 2014.: Keys to soil taxonomy, Soil Conserv. Serv., 12, 410. https:// doi.org/:10.1109/TIP.2005.854494, 2014.

Sonneveld, M. P. W., Bouma, J., and Veldkamp, A., 2002.: Refining soil survey information for a Dutch soil series using land use history, Soil Use Manag., 18,(3), 157–163, 2002.

Steduto, P., Hsiao, T.C., Raes, D., Fereres, E., 2009. AquaCrop-The FAO crop model to simulate yield response to water: I. Concepts and underlying principles. Agron. J. 101, 426–437.

Stöckle, C.O., Donatelli, M., Nelson, R., 2003. CropSyst, a cropping systems simulation model. Eur. J. Agron. 18, 289–307.

[revised manuscript text omitted]

* (average value ± standard deviation)

---

## Author Response (AR2)

Dear Editor, Thank you for your positive comments. We have done the requested revisions to the original manuscript. As regard the comment about the use of Soil Taxonomy system, we want to stress that we referred to the soil series concept (it is part of ST approach) and that it is widely accepted international system (it is adopted in several countries, as well as Italy where the work was carried out).

Thanks again

Sincerely yours, Antonello Bonfante, also on behalf of the co-authors.